# A Balanced Sowing Density Improves Quality of Rapeseed Blanket Seedling

**Qingsong Zuo** [1,*], **Jingjing You** [2,3,4], **Long Wang** [2,3,4], **Jingdong Zheng** [2,3,4], **Jing Li** [2,3,4], **Chen Qian** [2,3,4], **Guobing Lin** [2,3,4], **Guang Yang** [2,3,4] **and Suohu Leng** [2,3,4]

1   Agricultural College, Yangzhou University, 48 Wenhui East Road, Yangzhou 225009, China
2   Jiangsu Key Laboratory of Crop Genetics and Physiology, Yangzhou University, Yangzhou 225009, China; youjingjing1998@163.com (J.Y.); wanglong_yzu@163.com (L.W.); zjd19971014@163.com (J.Z.); mx120210720@yzu.edu.cn (J.L.); mz120211279@yzu.edu.cn (C.Q.); lgb1832659540@163.com (G.L.); yangguang@yzu.edu.cn (G.Y.); oilseed@yzu.edu.cn (S.L.)
3   Jiangsu Key Laboratory of Crop Cultivation and Physiology, Yangzhou University, Yangzhou 225009, China
4   Jiangsu Co-Innovation Center for Modern Production Technology of Grain Crops, Yangzhou University, Yangzhou 225009, China
*   Correspondence: qszuo@yzu.edu.cn

**Abstract:** Mechanized transplanting of rapeseed (*Brassica napus* L.) blanket seedling is an effective strategy to cope with the seasonal conflict and large labor cost in rapeseed production. The sowing density is a key factor to cultivate high-quality seedlings suitable for mechanized transplanting. An experiment was conducted to investigate the effects of different sowing density levels of 638, 696, 754, 812, 870 and 928 seeds per tray (referred as D1, D2, D3, D4, D5 and D6, respectively) on agronomic traits and survival rate after mechanized transplanting of two rapeseed cultivars (Zheyouza108 and Heza17) in 2020 and 2021. The results showed that high sowing density increased plant height but decreased leaf area, collar diameter, biomass accumulation, the ratio of root to shoot and seedling fullness. These negative effects jointly decreased the seedling rate and survival rate after mechanized transplanting. However, the seedlings under D1 and D2 posed a great plant survival rate of more than 95% after mechanized transplanting, suggesting that the seedlings under the two densities were perfect for mechanized transplanting. In addition, hierarchical analysis grouped D1 and D2 into the same class, indicating that their seedling qualities were not significantly different, though the blanket seedlings under D1 outperformed those under D2 in some traits. A sowing density of 696 seeds per tray (D2) is then recommended in this study, altogether considering its high-quality seedlings suitable for mechanized transplanting, and economically, fewer seedling trays required.

**Keywords:** rapeseed blanket seedling; mechanized transplanting; sowing density; survival rate

## 1. Introduction

Rapeseed (*Brassica napus* L.) is an important oil crop due to its high-quality edible oil for human consumption and rich protein source for livestock [1,2]. The oil derived from rapeseed has a high nutritional value with a favorable fatty-acid composition [3]. As a large producer in rapeseed production, China is in possession of about 7.0 million hectares of rapeseed planting area, which accounts for 23% of the total planting area around the world [4]. Rapeseed offers approximately 50% of the domestic vegetable oil in China. It is estimated that over 85% of the rapeseed planting area in China is located in the Yangtze River Basin where the winter rapeseed is grown [5].

In the Yangtze River Basin, both direct sowing and transplanting are used for establishing winter rapeseed [6]. Crop transplanting is easy to obtain suitable planting density and field microclimate [7,8], so the transplanting method is preferred in this area due to its improvements in plant growth and seed yield. More specifically, the direct-sowing method faces a severe seasonal conflict due to the late harvest of preceding crops such as rice,

resulting in shortened vegetative growth before winter and as well as the whole growth duration of winter rapeseed [9]. In addition, environmental stress such as seasonal drought and low temperature often occur in autumn and winter seasons in the Yangtze River Basin, which exerts negative effects on seed germination and leads to a weak seedling [10,11]. Compared with the direct-sowing method, the transplanting method can cultivate seedlings in the seedbed and then select the strongest to transplant in the main field. This practice reduces the growth duration in the main field, lessening the seasonal conflict. Moreover, the transplanting method can cultivate stronger seedlings with greater vigor and stronger tolerance through management practices that offer an opportunity to survive during the overwintering period and establish a great basis for subsequent reproductive growth [12]. It was reported that the transplanting method produces more biomass accumulation, higher seed yield and higher oil yield due to effective utilization of resources including nutrients, sunshine, water and temperature [13,14].

Traditionally, the rapeseed transplanting from seedbed to main field is completed manually. Therefore, the transplanting method is considered as a labor-intensive cultivation practice that requires a large amount of agricultural labors and significantly increases the cost of rapeseed production, especially in today's society engaging rural labors migrated to cities in China [15]. The mechanized transplanting is an effective strategy to decrease labor input and improve transplanting efficiency to secure winter rapeseed production. Since 2010, Yangzhou University and Nanjing Research Institute of Agricultural Mechanization, Ministry of Agriculture have jointly developed the key technical issues associated with mechanized transplanting of rapeseed blanket seedlings. There are two key factors gauging this novel technique, the special transplanting machine: one has been successfully developed, and cultivating high-quality rapeseed blanket seedlings for mechanized transplanting is the another.

Rapeseed blanket seedlings are characterized by high density, about 4000 seedlings per square meter. High density is beneficial to mechanized transplanting. However, high density often intensifies individual competition for nutrients, sunshine and water, and accelerates the plant growth, and ultimately forms weak seedlings that have difficulty surviving after transplanting and produce a high seed yield in the main field. It was demonstrated that high density in the seedbed can decrease seedling weight and stem thickness [16], limit leaf production such as leaf number and leaf area and develop a weak root system [17,18], and subsequently affect plant growth and yield formation [19]. Generally, a balanced density plays a critical role in the establishment of high-quality seedlings suitable for mechanized transplanting.

We hypothesized that an appropriate sowing density can cultivate high-quality blanket seedlings suitable for mechanized transplanting. As such, this study was conducted to explore the effects of sowing density on seedling agronomic traits and survival rate after mechanized transplanting.

## 2. Results

### 2.1. Seedling Number per Tray and Seedling Rate on the 30th Day after Sowing

The ANOVA results showed that sowing density and cultivar significantly affected seedling number per tray and seedling rate on the 30th day after sowing (Figure 1). The average seedling number per tray (Figure 1a,b), ranging from 614.3 to 810.2, was improved by high sowing density. The treatment of D6 produced the greatest value. The seedling number per tray under D6 treatment was increased by 30.39% for Zheyouza108 and 29.08% for Heza17 compared to D1 treatment, respectively. However, high sowing density showed a negative effect on the seedling rate (Figure 1c,d). For example, as compared with D1 treatment, the D6 treatment decreased the seedling rate by 10.36% for Zheyouza108 and 11.26% for Heza7, respectively. In addition, the seedling rate showed no significant difference among D1, D2 and D3 treatments.

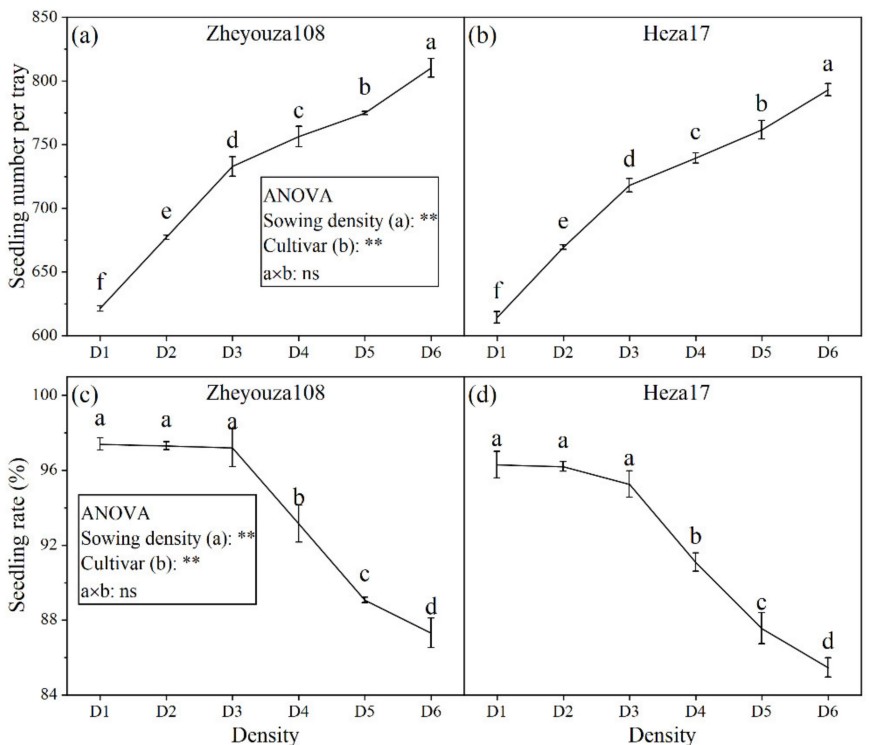

**Figure 1.** Effects of sowing density on seedling number per tray (**a**,**b**) and seedling rate (**c**,**d**) of rapeseed blanket seedling on the 30th after sowing of cultivar Zheyouza 108 (**a**,**c**) and Heza 17 (**b**,**d**). D1, D2, D3, D4, D5 and D6 represent sowing-density levels of 638, 696, 754, 812, 870 and 928 seeds per tray, respectively. Different letters indicate significant differences among treatments ($p \leq 0.05$). Probability levels are performed by ns and ** for not significant and 0.01, respectively.

### 2.2. Agronomic Traits on the 30th Day after Sowing

The ANOVA results (Table 1) showed that sowing density and cultivar interactively affected all agronomic traits on the 30th day after sowing, except for plant height.

**Table 1.** Effects of sowing density on agronomic traits of rapeseed blanket seedling on the 30th day after sowing.

| Cultivar | Sowing Density | Plant Height (cm) | Leaf Area (cm² per Plant) | Collar Diameter (mm) |
|---|---|---|---|---|
| Zheyouza108 | D1 | 8.84 f | 30.35 a | 2.30 a |
| | D2 | 9.02 e | 28.99 b | 2.21 b |
| | D3 | 9.31 d | 28.40 c | 2.15 c |
| | D4 | 9.64 c | 27.10 d | 2.01 d |
| | D5 | 9.95 b | 24.70 e | 1.85 e |
| | D6 | 10.24 a | 23.15 f | 1.72 f |
| Heza17 | D1 | 8.84 f | 28.79 a | 2.26 a |
| | D2 | 8.92 e | 27.02 b | 2.16 b |
| | D3 | 9.16 d | 26.91 b | 2.05 c |
| | D4 | 9.48 c | 25.63 c | 1.93 d |
| | D5 | 9.82 b | 24.22 d | 1.82 e |
| | D6 | 10.08 a | 22.23 e | 1.71 f |
| ANOVA | | | | |
| | Sowing density | ** | ** | ** |
| | Cultivar | ** | ** | ** |
| | Sowing density × Cultivar | ns | ** | ** |

D1, D2, D3, D4, D5 and D6 represent sowing-density levels of 638, 696, 754, 812, 870 and 928 seeds per tray, respectively. Different letters within a column indicate significant difference at $p \leq 0.05$ among the same cultivar. Probability levels are performed by ns and ** for not significant and 0.01, respectively.

The plant height (Table 1) ranged from 8.83 to 10.24 cm, and with the increasing sowing density, it was significantly increased. For example, plant height under D1 treatment was 15.92% for Zheyouza108 and 14.14% for Heza17, higher than this under D6 treatment, respectively. However, the change tendencies of leaf area and collar diameter (Table 1) differed from plant height. As sowing density increased, these two parameters decreased. The D6 treatment decreased leaf area and collar diameter by 23.71% and 25.24% for Zheyouza108, and 22.78%, respectively, and 24.23% for Heza17, respectively, as compared with D1 treatment.

*2.3. Biomass Weight on the 30th Day after Sowing*

The ANOVA results (Table 2) indicated that sowing density and cultivar significantly affected biomass weight, the ratio of root to shoot and seedling fullness on the 30th day after sowing. However, the interaction between them showed no significant effect.

The dry weight of root, shoot and whole plant (Table 2) decreased with the increased sowing density. More specially, the dry weight of root, shoot and whole plant were decreased by 29.60%, 20.14% and 21.03% for Zheyouza108, and 31.61%, 21.99% and 22.88% for Heza17, respectively, with the sowing density increasing from D1 to D6. Similarly, the ratio of root to shoot (Table 2), ranging from 8.85% to 10.30%, decreased with high density. This ratio was highest under D1 and D2 treatments, followed by D3 and D4 treatments, and the lowest one was produced in D6 treatment. The seedling fullness also significantly decreased with the increasing sowing density.

**Table 2.** Effects of sowing density on biomass accumulation of rapeseed blanket seedling on the 30th day after sowing.

| Cultivar | Sowing Density | Dry Weight (mg per Plant) | | | Ratio of Root to Shoot | Seedling Fullness (mg cm$^{-1}$) |
|---|---|---|---|---|---|---|
| | | **Whole Plant** | **Root** | **Shoot** | | |
| Zheyouza108 | D1 | 104.50 a | 9.76 a | 94.75 a | 0.103 a | 10.72 a |
| | D2 | 100.00 b | 9.27 b | 90.74 b | 0.102 a | 10.06 b |
| | D3 | 95.69 c | 8.56 c | 87.13 c | 0.098 b | 9.36 c |
| | D4 | 90.78 d | 8.01 d | 82.77 d | 0.097 b | 8.59 d |
| | D5 | 86.19 e | 7.39 e | 78.80 e | 0.094 c | 7.92 e |
| | D6 | 82.53 f | 6.87 f | 75.66 f | 0.091 d | 7.39 f |
| Heza17 | D1 | 103.05 a | 9.45 a | 93.60 a | 0.101 a | 10.60 a |
| | D2 | 98.36 b | 8.96 b | 89.40 b | 0.100 a | 10.02 b |
| | D3 | 92.83 c | 8.24 c | 84.58 c | 0.098 b | 9.23 c |
| | D4 | 87.93 d | 7.69 d | 80.24 d | 0.096 b | 8.46 d |
| | D5 | 83.90 e | 7.13 e | 76.77 e | 0.093 c | 7.82 e |
| | D6 | 79.47 f | 6.46 f | 73.01 f | 0.089 d | 7.24 f |
| ANOVA | | | | | | |
| Sowing density | | ** | ** | ** | ** | ** |
| Cultivar | | ** | ** | ** | ** | ** |
| Sowing density × Cultivar | | ns | ns | ns | ns | ns |

D1, D2, D3, D4, D5 and D6 represents sowing density levels of 638, 696, 754, 812, 870 and 928 seeds per tray, respectively. Different letters within a column indicate significant difference at $p \leq 0.05$ among the same cultivar. Probability levels are performed by ns and ** for not significant and 0.01, respectively.

*2.4. Survival Rate on the 10th Day after Mechanized Transplanting*

The ANOVA results showed that sowing density and cultivar significantly affected the survival rate on the 10th day after mechanized transplanting (Figure 2).

The survival rate after mechanized transplanting (Figure 2a,b) varied from 75.30% to 97.10% for Zheyouza108 and from 73.40% to 96.80% for Heza17. The seedling survival rates under D1 and D2 were consistently greater than 95%, whereas it decreased significantly with sowing density increasing from D3 to D6. As compared with D1 treatment, the D6 treatment decreased the survival rate after mechanized transplanting by 22.45% for Zheyouza108 and 24.17% for Heza17, respectively.

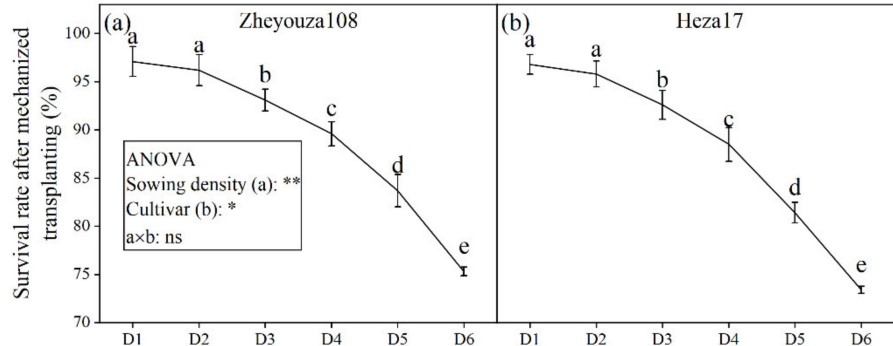

**Figure 2.** Effects of sowing density on survival rate on the 10th day after mechanized transplanting for cultivar Zheyouza 108 (**a**) and Heza 17 (**b**). D1, D2, D3, D4, D5 and D6 represent sowing density levels of 638, 696, 754, 812, 870 and 928 seeds per tray, respectively. Different letters indicate significant differences among treatments ($p \leq 0.05$). Probability levels are performed by ns, * and ** for not significant, 0.05 and 0.01, respectively.

### 2.5. Relationships

The results of correlation analysis (Figure 3) showed that the survival rates after mechanized transplanting were positively correlated with leaf area, collar diameter, dry weight of root, shoot and whole plant, the ratio of root to shoot and seedling fullness. However, it showed a negative relationship with plant height and seedling number per tray.

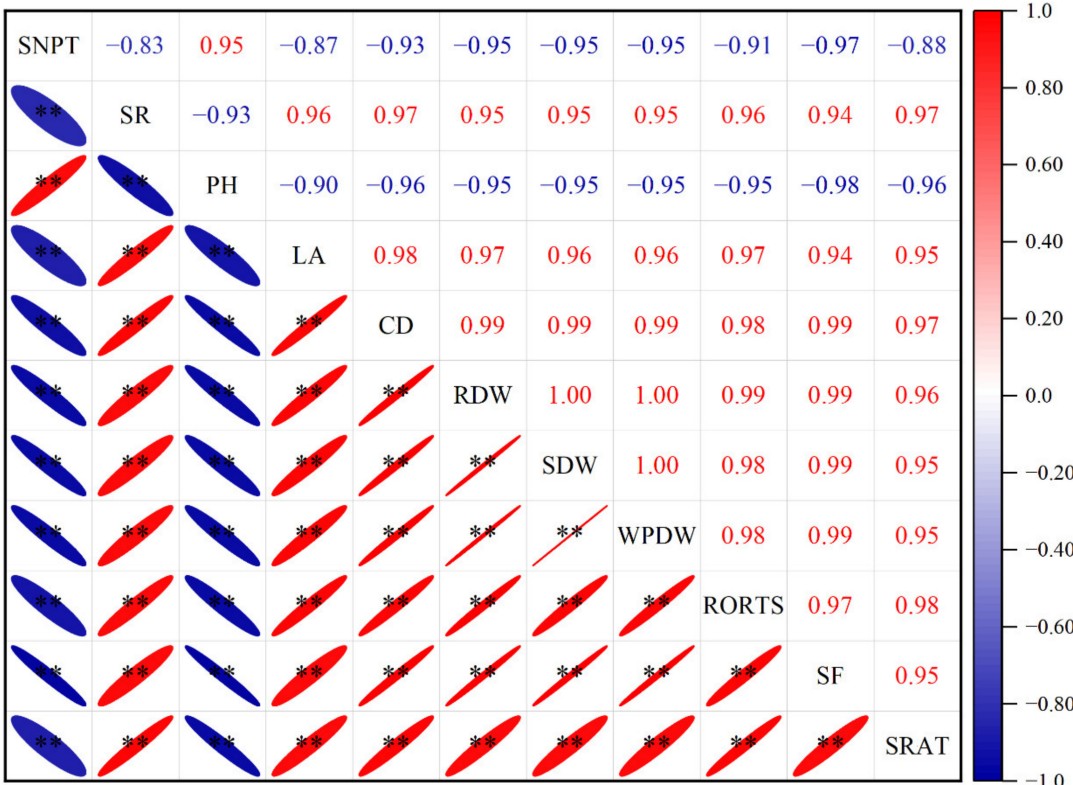

**Figure 3.** Pearson's correlation among different parameters in this study. SNPT: seedling number per tray; SR: seedling rate; PH: plant height; LA: leaf area; CD: collar diameter; RDW: root dry weight; SDW: shoot dry weight; WPDW: whole plant dry weight; RFRTS: ratio of root to shoot; SF: seedling fullness; SRAT: survival rate after mechanized transplanting. Probability levels are performed by ns and ** for not significant and 0.01, respectively.

Moreover, the heat map (Figure 4) showed a relation between treatments and studied parameters. The hierarchical analysis divided different treatments into two main groups (D1–D4 and D5–D6), and then D1–D4 treatments were divided into two sub-main groups (D1–D2 and D3–D4). These results suggested that D1 and D2 treatments showed the highest quality, supported with greater leaf area, collar diameter, biomass weight, ratio of root to shoot, seedling fullness and survival rate after mechanized transplanting, except for plant height and seedling number after sowing, followed by D3 and D4 treatments, and the lowest were D5 and D6 treatments.

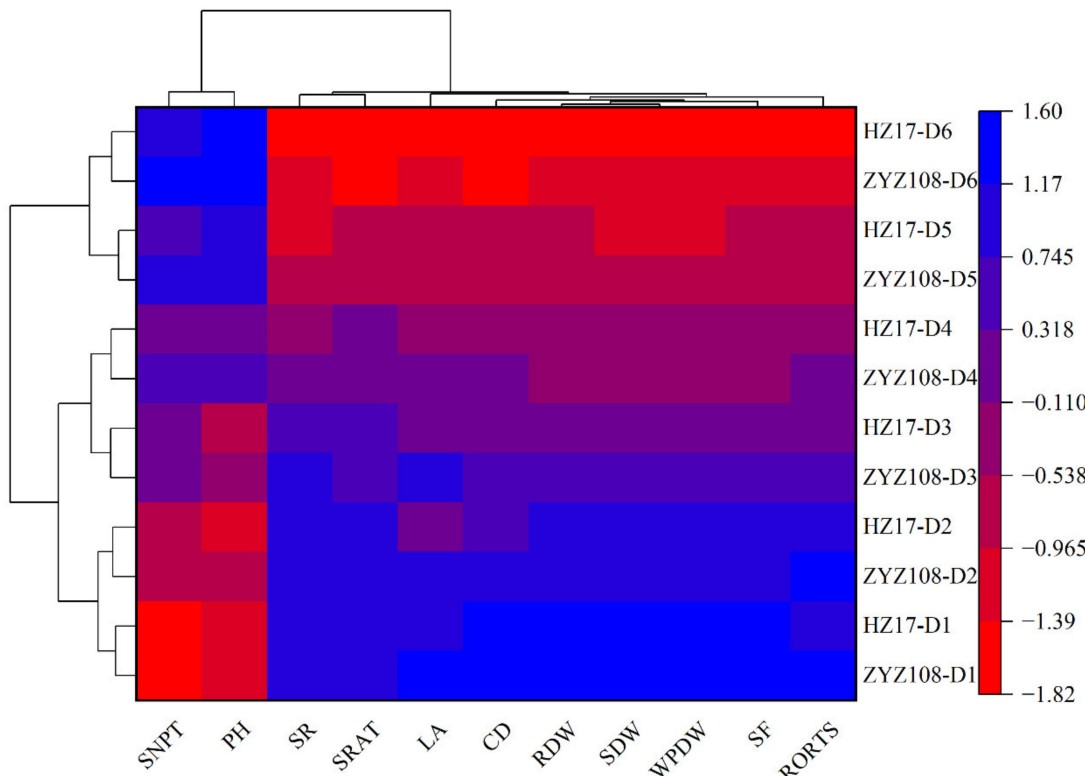

**Figure 4.** The hierarchical-clustering analysis between studied parameters and treatments. "Zheyouza108" and "Heza17" were abbreviated ZYZ108 and HZ17. D1, D2, D3, D4, D5 and D6 represent sowing-density levels of 638, 696, 754, 812, 870 and 928 seeds per tray, respectively. SNPT: seedling number per tray; SR: seedling rate; PH: plant height; LA: leaf area; CD: collar diameter; RDW: root dry weight; SDW: shoot dry weight; WPDW: whole plant dry weight; RFRTS: ratio of root to shoot; SF: seedling fullness; SRAT: survival rate after mechanized transplanting.

## 3. Discussion

Sowing density is an important factor affecting plant growth and development in field [20,21]. In our study, an increase in sowing density of blanket seedlings significantly increased plant height but decreased leaf area and biomass accumulation (Tables 1 and 2). Similar results were observed in rice, in that the seedlings became taller and lighter with less leaf and root growth due to high sowing density [22]. These results can be attributed to the fact that the high sowing density intensifies the individual competition for resources such as nutrients, water and sunshine, and thereafter limits individual growth [23]. Our study also showed the decreased seedling fullness under a high-sowing-density condition (Table 2). Increased plant height accompanied with decreased biomass accumulation can reduce the stem strength and increase the risk of lodging after transplanting [24], ultimately resulting in inhibited plant growth and decreased seed yield at the later vegetative and reproductive-growth stages [7]. As a significant indicator of seedling quality [25], the collar diameter was decreased with the increasing sowing density (Table 1). The thinner collar diameter leads blanket seedlings to risk in being broken during mechanized transplanting, and

seedlings with thinner collar diameter are difficult to grow upright after transplanting [26]. Conclusively, these adverse effects of high sowing density together result in the poor-quality seedling and thereafter decreased survival rate after mechanized transplanting, despite increased seedling number per tray (Figure 3).

Regarding biomass accumulation, high sowing density decreased the dry weight of root and shoot (Table 2). The inhibited root growth due to high sowing density decreases the ability of root systems to absorb water and nutrients from soil, and therefore limits the shoot growth [27]. Results from our study that the root dry weight was positively correlated with the shoot dry weight support this statement (Figure 3). Generally, plants exposed to environmental stress such as nutrient deficiency would increase the ratio of root to shoot, which improves the absorption capacity of root and promotes tolerance against stress [28,29]. However, in our study, a decreased ratio of root to shoot was recorded under the high sowing density (Table 2), suggesting that root growth is more sensitive to high sowing density than shoot growth. Under a high-density condition, the limited resources in the underground contribute to low individual allocation of resources absorbed through root systems [30]. In other word, the return on investment in root is low. Therefore, blanket seedlings put more investments in the shoot and shift the competition for resources from underground to aboveground, such as rich sunshine resource. This is a self-protective mechanism for blanket seedlings to maximize resource utilization under a high-density condition.

The density of rapeseed blanket seedling for mechanized transplanting is significantly higher than that of direct-seeding rapeseed, and the rapeseed blanket seedlings are relatively tenderer and weaker, so some seedlings may not survive after mechanized transplanting. According to our previous investigation, if the blanket seedlings did not grow upright within 10 days after mechanized transplanting, they would eventually die. Therefore, we usually investigate the seedling survival rate on the 10th day after mechanized transplanting. Previous research showed that the differences of the final seed yields of rapeseed blanket seedling between 95% survival rate and 100% survival rate were not significant [31]. If the survival rate decreased further, the seed yield at the maturing stage decreased significantly. In this study, as the sowing density increased, the survival rate after mechanized transplanting showed a decreasing trend. That may be attributed to poor agronomic traits including taller plant height, thinner collar diameter and less biomass accumulation (Figure 3). However, the survival rate after mechanized transplanting showed a stable trend and more than 95% under D1 and D2 treatments, indicating that the seedlings under 638 seeds per tray (D1) and 696 seeds per tray (D2) treatments were feasible to mechanized transplanting. Moreover, the hierarchical analysis divided D1 and D2 into the same group, suggesting that although the blanket seedlings under D1 treatment outperformed than those under D2 treatment, there was no significant difference between them (Figure 4). For the technology of mechanized transplanting of rapeseed blanket seedlings, we should first of all ensure a high survival rate after mechanized transplanting of seedlings. At the same time, high-density rapeseed blanket seedling is beneficial to reduce the cost of seedling raising. Therefore, the sowing density at 696 seeds per tray (D2) is recommended to cultivate the high-quality seedlings suitable for mechanized transplanting.

## 4. Materials and Methods

### 4.1. Experimental Set, Cultivar and Substrate

This experiment was conducted in the experimental farmland in Yangzhou University (32.30° N, 119.43° E), Jiangsu Province, China in 2020 and 2021. Two rapeseed cultivars of Zheyouza 108 and Heza 17, widely grown in this region, were used in this experiment. Zheyouza 108 and Heza 17 was cultivated by Zhejiang Academy of Agricultural Sciences, China and Shanghai Academy of Agricultural Sciences, China, respectively. The sowing date was 6 October 2020 and 8 October 2021, respectively. The substrate used to cultivate seedlings (produced by Jiangsu Xingnong Substrate Technology Co., Ltd., Zhenjiang, China) had total N, P and K nutrients of ≥3%, total organic matter of ≥35% and pH of 5.8–7.0. Di-

mensions of these plastic trays used in this experiment were 575 mm × 275 mm × 25 mm (length × width × height).

This experiment was arranged in a randomized block design with two cultivars and six sowing-density levels in three replications. The six density levels consisted of 638, 696, 754, 812, 870 and 928 seeds per tray (referred as D1, D2, D3, D4, D5 and D6, respectively). 20 trays were set for each treatment for sampling and mechanized transplanting. Different sowing-density levels were achieved by varying the number of rows of the self-designed seeder, with the fixed number of seeds per row (58 seeds per row).

The blanket seedlings in each tray were fertilized by dissolving 0.25 g and 1.5 g N per tray in 100 mL water to apply at thecotyledon stage and one-leaf and one-heart stage, respectively. In addition, 1.5 mg of uniconazole wettable powder (active ingredient content 5%) per tray was used at one-leaf and one-heart stage to secure the quality of blanket seedlings [31].

### 4.2. Sampling and Measurement

On the 30th day after sowing, the seedling number per tray and seedling rate were measured; the seedling rate was calculated by divided the seedling number per tray by sowing density. 20 plants for each treatment were sampled to measure these selected parameters such as plant height, leaf area and collar diameter. Then, these plants were divided into root and shoot, and dried at 80 °C and weighted. The ratio of root to shoot was calculated by dividing the root dry weight by shoot dry weight. The seedling fullness was measured to determine the strength of rapeseed blanket seedling and it was calculated by dividing the shoot dry weight by plant height.

At the time, blanket seedlings were mechanized-transplanted in field plots by the special transplanter for rapeseed blanket seedlings (Model 2ZYG-6). Each plot was 30 m × 2 m (length × width). Seedlings were irrigated in time after mechanized transplanting. According to our previous studies, if the seedlings did not grow upright within 10 days after mechanized transplanting, they would eventually die; therefore, the survival rate after mechanized transplanting was investigated on the 10th day after mechanized transplanting.

### 4.3. Statistical Analysis

The analysis of variance (ANOVA) was performed with SPSS statistical 20 software (SPSS Inc., Chicago, IL, USA), and the means were compared by the Duncan's multiple range test at the $p = 0.05$ level. Graphs, Pearson correlations and the heatmap were performed with Originpro. All the selected parameters were shown as average values of the two-year experiments because there was no significant difference between the two years.

## 5. Conclusions

In this study, the high sowing density significantly increased plant height but decreased leaf area, collar diameter and biomass accumulation, the ratio of root to shoot and seedling fullness. Therefore, the seedling rate and survival rate after mechanized transplanting decreased due to the poor quality of seedling under the high-density condition. The survival rates of rapeseed blanket seedling under 638 seeds-per-tray and 696 seeds-per-tray treatments were higher than 95%, so the seedlings under these two density treatments were feasible to mechanized transplanting. At the same time, high-density rapeseed blanket seedling is beneficial to reduce the cost of seedling raising. Therefore, the sowing density at 696 seeds per tray is recommended in this study, altogether considering that its high-quality seedlings are suitable for mechanized transplanting, and its lower seedling-raising cost.

**Author Contributions:** Q.Z., S.L. and G.Y. designed the research; L.W., J.Y., J.Z., J.L., C.Q. and G.L. performed the experiments and data collection; Q.Z., J.Y. and L.W. contributed to data analysis and wrote the paper. All authors have read and agreed to the published version of the manuscript.

**Funding:** This research was financially funded by the National Key Research and Development Program of China (2018YFD1000900), the Major Project of Basic Science (Natural Science) Research

in Colleges and Universities in Jiangsu Province, China (21KJA210003), and Demonstration and Promotion Project of Key Technologies for Improving Quality and Efficiency of Rapeseed, Yangzhou, China (YN202229).

**Institutional Review Board Statement:** Not applicable.

**Informed Consent Statement:** Not applicable.

**Data Availability Statement:** Not applicable.

**Conflicts of Interest:** The authors declare no conflict of interest.

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
