# Peer review of "A Balanced Sowing Density Improves Quality of Rapeseed Blanket Seedling"

_agronomy, doi:10.3390/agronomy12071539_

Round 1

Reviewer 1 Report

This is a well written article with an obvious agronomic significance for a very important field crop - rapeseed. However, with all the appreciation to the agronomic and technical achievements and contribution, this work does not fall within the scientific scope of "Plants". There is nothing new, scientifically, in the principal responses of seedlings density on their development, in spite of the significance to the rapeseed industry. Hence, unfortunately, I recommend to reject this article and to try submitting it to one of the agronomic journals.

Statistical design must be detailed further, in any case.

Author Response

Response:Thank the reviewer for their comments and suggestions.

Rapeseed is an important oil crop in the world. This paper studies the problem of density setting under a new cultivation mode, which belongs to the category of efficient cultivation, and this paper studies belong to the scientific scope of "Plants".

This new technology is relatively novel, and many people are interested in it. We applied for relevant patents in Canada. In 2016, Yanma Company of Japan purchased our patented technology. In the new efficient cultivation mode, the supporting cultivation techniques need to be carefully studied.

The statistical software used is also popular in this paper. In the experimental design, we detailed it and supplemented the detailed number of seedling trays for each treatment.

Reviewer 2 Report

Zuo etal investigated the effects of different sowing density levels on agronomic traits and survival rate after mechanized transplanting of two rapeseed cultivars, and recommended a specific seeding rate. My concern is that the experiment was not well designed to address the research questions. Two cultivars do not provide any statistically sound conclusion and the statistical analyses including two-way ANOVA (treatment by cultivar)  and correlation analysis do not seem plausible . It would be better if the authors increase the number of cultivars to reduce background noise,  including cultural practice as control and demonstrate economic benefits of the newly recommended seed rate. Otherwise, the experiment does not give any considerable evidence or add to something our existing knowledge. As we increase the density, the plants will be taller and weaker. As a result, we expected yield reduction.

Author Response

Response:Thank the reviewer for their comments and suggestions.

Generally speaking, in the research of high efficient cultivation, two or three representative cultivars can explain the problem. In this paper, these two varieties, Zheyouza 108 and Heza 17, were widely planted in this region.

Because of high density of blanket seedlings, the seedling is tender and weak, and the key is the survival rate after mechanized transplanting. As for the final yield, it mainly depends on the cultivation measures after mechanized transplanting.

Reviewer 3 Report

Specific comments:

Following points if deemed for incorporation may be quite useful for the authors, to increase the readability and more clear presentation of the manuscript:

Line 16-17: delete the double quote sign of the two cultivars, as the first letter of their names was already capitalized. Same on Line 233;

Line 19: changed the word “jointly” to “eventually”;

Line 20-26: re-write the sentences. “However, the seedlings under D1 and D2 posed a great plant survival rate of more than 95% after mechanized transplanting, suggesting the seedlings under the two densities were prefect for mechanized transplanting. In addition, the hierarchical analysis grouped D1 and D2 into same class, indicating their seedling qualities were not significantly different, though the blanket seedlings under D1 outperformed that under D2 in some traits. The sowing density at 696 plants per tray (D2) is then recommended in this study, all together considering its high-quality seedlings suitable for mechanized transplanting and economically less seedling blankets required”.

Line 43: change to “resulting in shorted vegetative growth before winter and as well as the whole growth duration of winter rapeseed”;

Line 49: change “ so avoid” to “lessening”;

Line 50: Change “greater growth” to “greater vigour”;

Line 59: change “ under the background of” to “in today’s society engaging”;

Line 64-67: re-write the sentences. “There are two key factors gauging this novel technique: the special transplanting machine, the one has been successfully developed and, cultivating high-quality rapeseed blanket seedlings for mechanized transplanting is the another”.

Line 69: change “High density is beneficial to mechanized transplanting” to” Generally, high density is economically beneficial to mechanized transplanting as less seedling blankets required”;

Line 71: grammar error. Change verbs of “accelerate” and “form” plural to their single;

Line 75 to 82: re-write and organize the sentence and paragraph. “We hypothesized that an appropriate sowing density can cultivate high-quality blanket seedlings suitable for mechanized transplanting. As such, this study was conducted to explore the effects of sowing density on seedling agronomic traits and survival rate after mechanized transplanting.”

Line 90: replace word ”than” with “ as compared to”;

Line 97 and Line 99: add word “ respectively” to correspond the relationship between the density levels (638, 696…) to their abbreviations (D1, D2…), as well as the significant levels to their symbols. Do the same thing in the same contexts throughout the paper, e.g. on Line 109-110, on other figure tiles and table footnotes;

Line 100-103: re-write the sentence. “The ANOVA results (Table 1) showed that sowing density and cultivar interactively affected all agronomic traits on the 30th day after sowing except for plant height”. Remember, if there is a significant interactive effect between two experimental factors (e.g. cultivar and density in this study), don’t separately explained their significant main effects as it is statistically wrong”;

Line 118: delete the last sentence.

Line 135-137: Re-write the sentence. “The seedling survival rates under D1 and D2 were consistently greater than 95%, whereas it decreased significantly with sowing density increasing from D3 to D6”.

Line 158: replace “including” with “ supported with greater”;

Line 15-160: delete “ except plant height and seedling number after sowing”;

Line 172-173: Question: The data showed the seedling quality under D1 and D2 was not significantly different, D1 even outperformed D2 in some traits, why the authors recommended D2 rather than D1? Economic reasons? Please explain here. Same on Line 226-228;

Line 225, Line 280: use more scientific sense “significant” instead of word “overall”;

Line 241: use “consisted of” instead of “included”;

Line 246: change “one leaf one heart” to “one-leaf and one-heart”.

Author Response

Thank the reviewer for their comments and suggestions. Thank the reviewer for doing a lot of revision work on the paper.

Following points if deemed for incorporation may be quite useful for the authors, to increase the readability and more clear presentation of the manuscript:

Line 16-17: delete the double quote sign of the two cultivars, as the first letter of their names was already capitalized. Same on Line 233;

Response: Done as requested.

Line 19: changed the word “jointly” to “eventually”;

Response: Done as requested.

Line 20-26: re-write the sentences. “However, the seedlings under D1 and D2 posed a great plant survival rate of more than 95% after mechanized transplanting, suggesting the seedlings under the two densities were prefect for mechanized transplanting. In addition, the hierarchical analysis grouped D1 and D2 into same class, indicating their seedling qualities were not significantly different, though the blanket seedlings under D1 outperformed that under D2 in some traits. The sowing density at 696 plants per tray (D2) is then recommended in this study, all together considering its high-quality seedlings suitable for mechanized transplanting and economically less seedling blankets required”.

Response: Done as requested.

Line 43: change to “resulting in shorted vegetative growth before winter and as well as the whole growth duration of winter rapeseed”;

Response: Done as requested.

Line 49: change “ so avoid” to “lessening”;

Response: Done as requested.

Line 50: Change “greater growth” to “greater vigour”;

Response: Done as requested.

Line 59: change “ under the background of” to “in today’s society engaging”;

Response: Done as requested.

Line 64-67: re-write the sentences. “There are two key factors gauging this novel technique: the special transplanting machine, the one has been successfully developed and, cultivating high-quality rapeseed blanket seedlings for mechanized transplanting is the another”.

Response: Done as requested.

Line 69: change “High density is beneficial to mechanized transplanting” to” Generally, high density is economically beneficial to mechanized transplanting as less seedling blankets required”;

Response: Done as requested.

Line 71: grammar error. Change verbs of “accelerate” and “form” plural to their single;

Response: Done as requested.

Line 75 to 82: re-write and organize the sentence and paragraph. “We hypothesized that an appropriate sowing density can cultivate high-quality blanket seedlings suitable for mechanized transplanting. As such, this study was conducted to explore the effects of sowing density on seedling agronomic traits and survival rate after mechanized transplanting.”

Response: Done as requested.

Line 90: replace word ”than” with “ as compared to”;

Response: Done as requested.

Line 97 and Line 99: add word “ respectively” to correspond the relationship between the density levels (638, 696…) to their abbreviations (D1, D2…), as well as the significant levels to their symbols. Do the same thing in the same contexts throughout the paper, e.g. on Line 109-110, on other figure tiles and table footnotes;

Response: Done as requested.

Line 100-103: re-write the sentence. “The ANOVA results (Table 1) showed that sowing density and cultivar interactively affected all agronomic traits on the 30th day after sowing except for plant height”. Remember, if there is a significant interactive effect between two experimental factors (e.g. cultivar and density in this study), don’t separately explained their significant main effects as it is statistically wrong”;

Response: Done as requested.

Line 118: delete the last sentence.

Response: Done as requested.

Line 135-137: Re-write the sentence. “The seedling survival rates under D1 and D2 were consistently greater than 95%, whereas it decreased significantly with sowing density increasing from D3 to D6”.

Response: Done as requested.

Line 158: replace “including” with “ supported with greater”;

Response: Done as requested.

Line 15-160: delete “ except plant height and seedling number after sowing”;

Response: Done as requested.

Line 172-173: Question: The data showed the seedling quality under D1 and D2 was not significantly different, D1 even outperformed D2 in some traits, why the authors recommended D2 rather than D1? Economic reasons? Please explain here. Same on Line 226-228;

Response: This question is very meaningful. The density of rapeseed blanket seedling for mechanized transplanting is significantly higher than that of direct-seeding rapeseed, and the rapeseed blanket seedlings are relatively tenderer and weaker, so some seedlings may not survive after mechanized transplanting. In this paper, we selected the survival tree rate after mechanized transplanting as an entry point for research. Under the condition of high survival rate, high density rapeseed blanket seedling can reduce the cost of seedling raising. Therefore, the sowing density at 696 seeds per tray (D2) is finally recommended. In the discussion section, we have made many modifications and adjustments.

Line 225, Line 280: use more scientific sense “significant” instead of word “overall”;

Response: Done as requested.

Line 241: use “consisted of” instead of “included”;

Response: Done as requested.

Line 246: change “one leaf one heart” to “one-leaf and one-heart”.

Response: Done as requested.

Reviewer 4 Report

This is an interesting research paper regarding a balanced sowing density improves quality of rapeseed blanket seedling. Rapeseed is the most important oil crop in China. The per capita cultivated land is small and the multiple crop index is high in China. In the main production area of winter rapeseed in the Yangtze River Basin, there are obvious seasonal conflicts for rapeseed production because of the late harvesting time of previous crops. Mechanized transplanting of blanket seedling can alleviate the seasonal contradiction and this cultivation mode is very meaningful. In the process of rapeseed blanket seedling cultivation, density is the key factor affecting seedling quality. The results provided novel understandings on suitable density, agronomic traits and survival rate.

This paper is worth considering for publishing in the Journal of Plants. There are several minor changes as follows:

1 Page 3 “2.1. Seedling number per tray and seedling rate on the 30th day after sowing  …The seedling number per tray (Figure1a-b), ranged from 614.33 to 810.17…”. These values are the average of three replications, and reserving a decimal fraction in the average seedling number value is enough. This sentence should be changed to “The average seedling number per tray (Figure1a-b), ranged from 614.3 to 810.2”.

2 Page 6 “4.1. Experimental set, cultivar and substrate.  …This experiment was conducted in the experimental farmland in Yangzhou University (32.30° N, 119.43° E), Jiangsu Province, China in October 2020 and 2021 This experiment was conducted in the experimental farmland in Yangzhou University (32.30° N, 119.43° E), Jiangsu Province, China in October 2020 and 2021….” The detailed sowing date should be presented.

3 Page 6 “4.1. Experimental set, cultivar and substrate. …1.5 mg of uniconazole wettable powder (active ingredient content 5%) per tray was used…”. Uniconazole is a good growth regulator. In this paper, how to determine the dosage of uniconazole and what is the basis?

4 Page 6 “4.2. Sampling and Measurement”. Why choose 30 day seedling age for mechanized-transplanting?

5 Page 7 “5. Conclusion …the seedlings under D1 and D2 treatments were feasible to mechanized transplanting… ” D1and D2 should be replaced by concrete treatment.

6 Page 14 and 15 “Table 1 and Table 2”. “Sowing density*Cultivar” Should be changed to “Sowing density×Cultivar”.

Author Response

1 Page 3 “2.1. Seedling number per tray and seedling rate on the 30th day after sowing  …The seedling number per tray (Figure1a-b), ranged from 614.33 to 810.17…”. These values are the average of three replications, and reserving a decimal fraction in the average seedling number value is enough. This sentence should be changed to “The average seedling number per tray (Figure1a-b), ranged from 614.3 to 810.2”.

Response: Done as requested.

2 Page 6 “4.1. Experimental set, cultivar and substrate.  …This experiment was conducted in the experimental farmland in Yangzhou University (32.30° N, 119.43° E), Jiangsu Province, China in October 2020 and 2021 This experiment was conducted in the experimental farmland in Yangzhou University (32.30° N, 119.43° E), Jiangsu Province, China in October 2020 and 2021….” The detailed sowing date should be presented.

Response: Done as requested. In this paper, we added the sowing date (The sowing date was October 6, 2020 and October 8, 2021, respectively.)

3 Page 6 “4.1. Experimental set, cultivar and substrate. …1.5 mg of uniconazole wettable powder (active ingredient content 5%) per tray was used…”. Uniconazole is a good growth regulator. In this paper, how to determine the dosage of uniconazole and what is the basis?

Response: As for the dosage of uniconazole, we have completed relevant research in the past two years and proposed that 1.0 and 1.5 mg per tray are feasible. In this paper, we marked the relevant references.

4 Page 6 “4.2. Sampling and Measurement”. Why choose 30 day seedling age for mechanized-transplanting?

Response: This choice is mainly based on experimental practice. Too small seedlings can not be transplanted by machine, and too large seedlings for mechanized transplanting is uneconomical.

5 Page 7 “5. Conclusion …the seedlings under D1 and D2 treatments were feasible to mechanized transplanting… ” D1and D2 should be replaced by concrete treatment.

Response: Done as requested.

6 Page 14 and 15 “Table 1 and Table 2”. “Sowing density*Cultivar” Should be changed to “Sowing density×Cultivar”.

Response: Done as requested.

Round 2

Reviewer 1 Report

The paper is good, scientifically as well as technically. I insist, however, that it belongs to the agronomic field of publications rather than to Plants. The paper does not include sufficient information regarding the physiological roots that lead to the displayed agronomic results. 

Author Response

Thank the reviewer for comments and suggestions.

At present, this research is mostly about agronomic traits. We also believe that with the further application and promotion of this new technology, more in-depth research will be carried out.

Reviewer 2 Report

I believe the study mus be improved by designing the experiment in a way to answer the basic economic and biological question of  plant density. In the current form, it does not add any layer of scientific knowledge. 

Author Response

Thank the reviewer for comments and suggestions.

The density of rapeseed blanket seedling for mechanized transplanting is significantly higher than that of direct-seeding rapeseed, and the rapeseed blanket seedlings are relatively tenderer and weaker, so some seedlings may not survive after mechanized transplanting. In this paper, we selected the survival rate after mechanized transplanting as an entry point for research. The focus of this article is on the effect of density on agronomic characteristics. The results showed that the seedlings under D1 and D2 posed a great plant survival rate of more than 95% after mechanized transplanting, and the hierarchical analysis grouped D1 and D2 into the same class. In the conclusion, we recommended the sowing density at 696 seeds per tray (D2) considering low cost.

In this paper, we studied the effects of density on agronomic characteristics of rapeseed blanket seedling by a randomized block design. This is also a conventional experimental design. The original intention of this article was not to study its economic benefits. Under the condition of meeting agronomic parameters, economic benefit is only an incidental secondary factor. In the discussion part of the last revised manuscript, we have made many modifications and adjustments.

The reviewer suggested to design an experiment to answer the basic economic and biological question. We feel it too difficult.